# Cerebrospinal and Brain Proteins Implicated in Neuropsychiatric and Risk Factor Traits: Evidence from Mendelian Randomization

**DOI:** 10.3390/biomedicines12020327

**Published:** 2024-01-31

**Authors:** Roxane de La Harpe, Loukas Zagkos, Dipender Gill, Héléne T. Cronjé, Ville Karhunen

**Affiliations:** 1Unit of Internal Medicine, Department of Medicine, University Hospital of Lausanne, 1011 Lausanne, Switzerland; 2Department of Epidemiology and Biostatistics, School of Public Health, Imperial College London, London SW7 2BX, UK; l.zagkos@imperial.ac.uk (L.Z.); dipender.gill@imperial.ac.uk (D.G.); 3Department of Public Health, Section of Epidemiology, University of Copenhagen, 1165 Copenhagen, Denmark; toinet.cronje@sund.ku.dk; 4Research Unit of Mathematical Sciences, Faculty of Science, University of Oulu, Fi-900014 Oulu, Finland; ville.karhunen@oulu.fi; 5Research Unit of Population Health, Faculty of Medicine, University of Oulu, Fi-900014 Oulu, Finland

**Keywords:** neuropsychiatric traits, genetically predicted proteins, cerebrospinal fluid proteins, brain proteins, plasma proteins, brain gene expression proteins, Mendelian randomization, causal inference, tissue-specific proteomic target

## Abstract

Neuropsychiatric disorders present a global health challenge, necessitating an understanding of their molecular mechanisms for therapeutic development. Using Mendelian randomization (MR) analysis, this study explored associations between genetically predicted levels of 173 proteins in cerebrospinal fluid (CSF) and 25 in the brain with 14 neuropsychiatric disorders and risk factors. Follow-up analyses assessed consistency across plasma protein levels and gene expression in various brain regions. Proteins were instrumented using tissue-specific genetic variants, and colocalization analysis confirmed unbiased gene variants. Consistent MR and colocalization evidence revealed that lower cortical expression of low-density lipoprotein receptor-related protein 8, coupled higher abundance in the CSF and plasma, associated with lower fluid intelligence scores and decreased bipolar disorder risk. Additionally, elevated apolipoprotein-E2 and hepatocyte growth factor-like protein in the CSF and brain were related to reduced leisure screen time and lower odds of physical activity, respectively. Furthermore, elevated CSF soluble tyrosine-protein kinase receptor 1 level increased liability to attention deficit hyperactivity disorder and schizophrenia alongside lower fluid intelligence scores. This research provides genetic evidence supporting novel tissue-specific proteomic targets for neuropsychiatric disorders and their risk factors. Further exploration is necessary to understand the underlying biological mechanisms and assess their potential for therapeutic intervention.

## 1. Introduction

Mental disorders affect one in every eight people around the world and represent a substantial burden on public health. Approximately 5% of the global years of life lost to disability are accounted for by mental disorders, with more than 125 million years lost in 2019 alone [1]. The development of effective therapies for mental disorders is challenging due to their complexity and heterogeneous aetiology and clinical presentation [2]. Liability to these disorders is determined by a complex interaction of distinct and shared genetic, lifestyle, and environmental factors. Disentangling these disorders is further complicated by the fact that risk factors of one disorder can often be the consequence of another or of its therapy (e.g., disturbed sleep) [3,4].

Proteins represent intermediate phenotypes for health outcomes and can provide insight into how genetic and non-genetic factors are mechanistically linked to clinical outcomes. Therefore, circulating proteins, which are measurable and easily accessible, are promising targets for diagnosis, prognosis, and intervention [2]. Moreover, proteomic analysis of specific tissues can reveal changes in protein expression patterns pertinent to the pathophysiology of the relevant organ system.

Genome-wide association studies (GWAS) have discovered numerous genetic variants predicting susceptibility to neuropsychiatric traits [5]. Approximately 90% of these variants are located outside of protein-coding regions, indicating that a substantial number of the associated variants may exert their influence on disease phenotypes through gene regulatory mechanisms, such as by influencing gene and protein expression [6,7,8]. Thus, by investigating the genetic variants that affect gene and protein expression, we will be able to more precisely unravel disease mechanisms and identify feasible therapeutic targets [6,7,9].

Performing such investigations in an epidemiological research setting can be tremendously challenging due to the biases of traditional observational studies, including challenges in sampling conditions, measurement of protein levels, and accounting for confounding factors like socioeconomic variables. Additionally, observed associations may be influenced by reverse causation, where neuropsychiatric or behavioural traits affect behaviours and biological pathways that, in turn, impact circulating protein levels [2]. Mendelian randomization (MR), an analytical framework that uses genetic variants as instrumental variables for studying the effect of varying an exposure on an outcome, can help to facilitate the inference of causal relationships. Genetic variants are assigned randomly during meiosis independently of environmental confounders and are fixed at conception, and thus not affected by outcomes, bypassing the limitations of traditional methods and in some ways mimicking a randomised controlled trial design [10]. 

In the current study, we aimed to determine the effect of protein abundance in the brain and cerebrospinal fluid (CSF) on the liability to neuropsychiatric disorders and their behavioural risk factors using the MR framework. Additionally, we assessed whether the effect of these proteins extended to their abundance in plasma and their region-specific expression in the brain.

## 2. Materials and Methods

### 2.1. Study Overview

Figure 1 provides an overview of the study design. Briefly, we identified genetic instruments for the relative abundance of 179 CSF and brain proteins and performed a two-sample cis-MR analysis to assess their causal effects on seven neuropsychiatric disorders [anorexia nervosa, attention deficit hyperactivity disorder (ADHD), autism spectrum disorder (ASD), bipolar disorder (BPD), insomnia, major depressive disorder (MDD), and schizophrenia], as well as 7 commonly associated behavioural risk factors of these disorders (alcohol consumption, fluid intelligence, educational attainment, physical activity, sleep duration, smoking, and leisure screen time). Then, to further explore the robustness of the associations, we followed up all statistically significant CSF and brain protein–outcome association pairs with a two-sample cis-MR of their concordant circulating protein levels and gene expression levels in the brain. Publicly available GWAS summary statistics were used for all analyses. Appropriate informed participant consent and ethical approval were obtained in each of the original studies.

### 2.2. Data Sources and Instrument Selection

#### 2.2.1. CSF and Brain Proteins

Genetic association data for 713 CSF and 1079 brain proteins were obtained from the National Institute on Aging Genetics of Alzheimer’s Disease Data Storage Site, repository at https://www.niagads.org/datasets/ng00102, (accessed on 30 March 2023). The data source included proteins that were assayed in CSF samples from 835 individuals (mean ± standard deviation [SD] age of 69.4 ± 9.3 years), including 621 without clinically diagnosed dementia. Additionally, proteomic analysis of brain tissue was performed in samples donated by 380 independent individuals (83.3 ± 10 years), including 44 without clinical dementia. Both cohorts represented individuals of European ancestry only, and both comprised more women than men (53% and 57%, respectively) [11].

Protein abundance was quantified in relative fluorescence units (RFUs), using a multiplexed aptamer-based platform developed by SomaLogic Inc. (Boulder, CO, USA) [12]. Genetic association analyses with protein abundance included sex, age, the first two genetic principal components, and the genotyping platform as covariates.

To select instruments, we considered variants associated with protein levels at genome-wide significance (*p* < 5 × 10^−8^) that were located within 1 Mb of the start and end coordinates of the corresponding gene based on the hg19/GRCh37 assembly coordinates (i.e., cis protein quantitative trait loci [pQTLs]) [13]. To ensure the independence of the genetic instruments, we clumped the variants using a pair-wise linkage disequilibrium (LD) r^2^ < 0.01 from the 1000 genomes project phase 3 European LD reference panel [14] within a clumping window of 1 Mb. In all cases, we used variants that were also present in the outcome datasets. Instrument strength was assessed using the proportion of variance in protein level, as well as individual and cumulative instrumental variant F-statistics, where F > 20 was considered acceptable [15]. Harmonisation of the datasets and clumping of the genetic instruments were performed using the “TwosampleMR” v.0.6.0 R package [16]. Details of the genetic variants used as instrumental variables are provided in Appendix A.

#### 2.2.2. Neuropsychiatric Diagnoses and Risk Factors 

Appendix A provides an overview of the outcome data sources used in this study. Briefly, mental disorders, physical activity, and insomnia were investigated as binary outcomes in their respective GWASs; i.e., having vs. not having a diagnosis of anorexia nervosa, ADHD, ASD, BPD, insomnia, MDD, or schizophrenia, and the presence vs. absence of moderate-to-vigorous intensity physical activity. Cohorts included at least 16,900 cases and 35,100 controls. 

Alcohol consumption was quantified as log_10_ transformed grams per day, while smoking was represented as a standardised smoking index that amalgamated smoking frequency and duration over the lifetime. Sleep duration and screen time were both investigated as hours per day. Education was measured as years of schooling completed, and fluid intelligence as the score obtained on a fluid intelligence test (SD units). Cohort sizes ranged from 257,828 to 766,345 participants. Except for the GWAS of schizophrenia, which was trans-ethnic (74.3% European), all other GWASs were performed in European cohorts.

#### 2.2.3. Mendelian Randomization: CSF and Brain Protein Levels 

We estimated the MR effect for each exposure–outcome pair using the Wald ratio (WR) when the instrument comprised a single variant, and the random-effects inverse-variance weighted method (IVW) when multiple variants were available as instruments [17]. MR estimates were reported per 1-log higher genetically predicted protein-binding aptamer relative fluorescence units (RFU). To correct for multiple testing of multiple correlated phenotypes, we calculated the false discovery rate (FDR) corrected *p*-values (pFDR). If more than 2 variants were available, we conducted MR-Egger [18] and MR-weighted median methods [19] to assess the robustness of the MR estimates to potential inclusion of pleiotropic variants. These methods have less statistical power to detect associations and were only applied as sensitivity analyses for the concordance of the effect sizes with the IVW method, and therefore no multiple testing correction was applied for the results from these methods. Last, to quantify heterogeneity, which may be an indication of horizontal pleiotropy [20], we calculated the I^2^ statistic.

#### 2.2.4. Follow-Up Analyses Using Plasma Protein Levels and Brain Gene Expression Data

For the exposure–outcome pairs that showed evidence of an MR association at pFDR < 0.05, we conducted follow-up MR analysis using genetic instruments of circulating plasma protein levels (pQTLs) and brain gene expression levels (eQTLs) where available. The genetic associations with plasma protein levels were obtained from genome-wide association studies conducted in a cohort of 35,559 Icelandic individuals [9]. Genetic associations with gene expression levels in the cortex, hippocampus, and spinal cord were obtained from a meta-analysis of 14 cohorts, consisting of up to 2683, 168, and 108 European ancestry individuals, respectively [21].

Instrument selection was done as described above; i.e., variants associated with protein abundance or expression at *p* < 5 × 10^−8^ within ±1 Mb of the coding gene, clumped at r^2^ < 0.01, were selected as instruments. To account for multiple testing, the pFDR values were calculated separately for the plasma protein and tissue-specific analyses. MR analyses were performed in the same manner as described for the primary analysis. MR estimates are reported per 1-SD increase for both circulating plasma protein-binding aptamer RFUs (effect sizes were calculated after inverse-normal rank transformations) and brain gene expression (effect sizes were calculated from z-scores assuming that var(y) = 1) and were considered statistically significant at pFDR < 0.05.

#### 2.2.5. Colocalisation Analyses

We performed colocalisation analyses to explore whether the associations observed between the two traits in MR were influenced by confounding byLD (i.e., at the same locus, there are genetic variants that influence the protein and outcome through separate biological mechanism but are liked to each other through LD) [22]. We assessed associations that were significant in both CSF and brain samples and those that were still at pFDR < 0.05 after the follow-up analyses. When using the Bayesian test for colocalization, we operated under the assumption that, at most, one causal variant for a trait existed within the locus. Under this assumption, this method was used to systematically test the following hypotheses: H0: no causal effects of variants with either (i.e., the exposure and the outcome) trait; H1: a causal variant–exposure effect only; H2: a causal variant–outcome effect only; H3: independent causal variant effects on the exposure and the outcome; and H4: a causal variant affecting both the exposure and outcome [23]. We used the default priors of 10^−4^, 10^−4^ and 10^−5^ for a variant being associated with exposure, outcome, and both traits, respectively [24]. A posterior probability (PP) for H4 (PPH4)  >  0.5 would imply that colocalization is more probable than any alternative scenario. Conversely, PPH3  >  0.5 would indicate that the observed MR associations are likely confounded by LD, and that there are distinct causal variants driving the instrument–exposure and instrument–outcome associations. Full summary statistics were required for these analyses. 

## 3. Results

### 3.1. CSF and Brain Genetic Instruments and Two-Sample MR Associations

A total of 173 CSF and 25 brain proteins were robustly instrumented by cis-pQTLs [median F-statistic = 82, interquartile range (IQR = 41–223) for CSF, and 67 (50–123) for brain instruments] and leveraged for two-sample MR analyses (Supplemental Appendix A). In total, 19 proteins were instrumented in both tissues (Figure 1C). 

We observed 77 associations at pFDR < 0.05 among 42 distinct instrumented CSF proteins across 11 outcomes (Appendix A), and 12 associations among eight distinct instrumented brain proteins across eight outcomes (Appendix A and Figure 1C). Of the 19 proteins with genetic instruments in both the CSF and brain, three were associated with at least one outcome in one tissue only (brain-derived PPAC with sleep duration, brain-derived C4b with fluid intelligence score and schizophrenia liability, and CSF-derived CNTN2 with schizophrenia liability), four (MSP, ILT-2, GSTP1, and ApoE2) had associations consistent in the brain and CSF (Table 1), and 12 did not associate with any outcome in either tissue. 

### 3.2. Protein-Specific and Outcome-Specific Associations in CSF/Brain Sample

Hepatocyte growth factor-like protein (MSP) showed the most extensive associations with our outcomes of interest. Higher genetically predicted relative abundance in the brain and CSF was associated with lower fluid intelligence test scores, lower odds of being physically active, prolonged screen time, and an increase in anorexia nervosa liability. Genetically predicted low-density lipoprotein receptor-related protein 8 (LRP8) abundance in the CSF had the second-broadest effect through its association with lower fluid intelligence scores, reduced odds of physical activity as well as a reduced liability to BPD and schizophrenia. Hours spent on leisure screen time emerged as the most widely associated trait with genetically predicted protein abundance, with 20 CSF protein-associations and two (overlapping) brain protein associations observed. Insomnia liability followed with 13 associations observed with instrumented CSF proteins, and three with proteins instrumented using abundance in the brain. Notably, apart from three proteins [one protein instrumented in each tissue (FCG2A and FCG2B) and one instrumented in both (GSTP1)], all the proteins associated with insomnia also had concordant associations with leisure screen time (Figure 2).

### 3.3. Plasma and Brain Gene Expression Genetic Instruments and Two-Sample MR Associations

Among the 46 CSF- and/or brain-derived proteins that showed significant associations, 27 were further investigated using their genetically determined abundance in plasma (Appendix A), while 28 (including 22 also tested in plasma) were investigated using their expression in brain tissue (i.e., gene expression in the cerebral cortex, spinal cord, and/or hippocampus, Appendix A). Of the 22 proteins with genetic instruments for both plasma abundance and brain expression level, three (LRP8, s-Tie-1 and MSP) had concordant associations when instrumented using either tissue. Eight proteins were associated only in one of the instrument modalities (Appendix A). 

Genetically predicted plasma abundance of six proteins and brain expression levels of seven proteins were each significantly associated with one or more neuropsychiatric disorder(s) or risk factor(s) (Table 1). All 12 associations implicating circulating proteins were directionally consistent to the associations observed for their CSF-derived counterparts. 

Genetically increased expression of CPNE1 protein in the spinal cord and hippocampus related to a lower fluid intelligence score, in line with observations in overall brain CPNE1 abundance. Our observation of higher instrumented CKAP2 abundance in the CSF increasing screen time also extended to *CKAP2* gene expression in the spinal cord and hippocampus.

Once again, LRP8 stood out as the protein with the broadest associations with the outcomes. Concordant directions of effect were observed between plasma- and CSF-derived relative LRP8 abundance, with higher LRP8 associated with lower fluid intelligence scores and odds of physical activity, as well as a reduced liability to BPD and schizophrenia. In contrast, the associations were opposite in the cerebral cortex. 

### 3.4. Evidence of Shared Causal Variants in Concordant Specific-Tissue Protein–Outcome Associations

Colocalisation results provided evidence for a shared causal variant between LRP8 abundance in the CSF and plasma and *LRP8* expression in the cerebral cortex and increased liability to BPD (PPH4  =  0.91 for all three), as well as fluid intelligence test scores (PPH4  =  0.70, 0.54, and 0.59, respectively). There was no evidence for a shared causal pathway between plasma LRP8 abundance and physical activity. There was strong evidence for a shared causal variant between CSF-derived sTie-1 and decreased fluid intelligence (PPH4 = 0.95), as well as increased liability to ADHD (PPH4 = 0.95) and schizophrenia (PPH4 = 0.90), but no evidence for sTie-1 abundance in the plasma or expression in the cerebral cortex. Finally, there was also consistent evidence that higher levels of ApoE2 and MSP in the CSF and brain relate to reduced leisure screen time and lower odds of engaging in physical activity, respectively (PPH4 > 0.90) (Table 2).

## 4. Discussion

### 4.1. Genetic Exploration of Neuropsychiatric Disorders: Unravelling Causal Tissue-Specific Protein Associations

This research provides genetic evidence in support of novel tissue-specific protein and gene targets for neuropsychiatric disorders and their risk factors. Using the MR framework, we provided evidence for a potential causal association between the genetically predicted relative abundance of 46 distinct proteins in the CSF and/or brain, across seven neuropsychiatric disorders (anorexia nervosa, ADHD, ASD, BPD, insomnia, MDD, and schizophrenia) and four of their commonly associated risk factors (fluid intelligence scores, participation in leisure physical activity, sleep duration, and screen time). We did not observe evidence for associations between genetically predicted protein abundance on lifetime smoking behaviour, alcohol consumption, or years of education. Association patterns observed among instrumented CSF proteins were largely concordant with those in the brain, with only three of the 19 proteins tested in both tissues not overlapping in their effects. A total of 33 proteins were successfully instrumented for their levels in plasma, or their expression in the cerebral cortex, spinal cord, and/or hippocampus. Twelve associations implicating genetically predicted plasma proteins, and seven implicating protein expression in the brain, were concordant to what was observed in our primary (CSF- and brain-derived protein levels) analyses. Six associations implicating gene expression in the brain opposed those observed for their corresponding protein levels in the brain or CSF, indicating a potential lack of connection between gene transcription and translation, or the involvement of negative feedback loops aimed at functional protein level regulation [25]. Alternatively, circulating proteins may affect CSF protein availability independent of their expression in the brain. Notably, the majority of CSF and brain pQTL–outcome pairs appear tissue-specific, confirming that certain genetic variants regulate the central nervous system differently from the periphery [9,25,26].

Overall, our results provided evidence to support a potential causal relationship for 16 distinct genetically predicted proteins that affect a similar outcome depending on their abundance in at least two different tissues. Our discussion focusses on the wider evidence of four of the proteins reported in our analyses. 

### 4.2. Focus on Specific Proteins: Insights into sTie-1, LRP8, ApoE2, and MSP

#### 4.2.1. Soluble Tyrosine-Protein Kinase Receptor Tie-1 (sTie-1)

We replicated findings from Gu et al. that higher genetically predicted sTie-1 in the CSF relates to higher schizophrenia liability [27], and from Lu et al. regarding the plasma abundance of sTie-1 and schizophrenia and ADHD liability [2], and extended these through analyses that identified only Tie-1_CSF_ to share a causal genetic variant with these outcomes. We also reported the novel association and colocalization of Tie-1_CSF_ and physical activity. *TIE1* expression in the dorsolateral prefrontal cortex was previously identified as one of nine differentially transcribed genes in a transcriptome-wide association study of ADHD (N = 19,099 cases) [28]. This finding was later replicated independently [29]. We reported similar *TIE1*_Cortex_-ADHD MR findings but did not identify evidence supporting a shared causal variant in this case. 

Our findings of a potentially causal association between CSF Tie-1 abundance and cortical expression and lower fluid intelligence scores reinforces the possibility of a causal underlying biological mechanism, given that certain studies have shown intelligence to be a protective factor in the development of ADHD and schizophrenia [30,31]. Other existing literature offers limited insights into the relationship between this protein, known to modulate TEK/TIE2 activity influencing angiogenesis regulation [32], and neurocognitive disorders. However, the tyrosine kinase family receptors have been studied in Alzheimer’s disease, wherein the soluble ectodomain of AXL receptor tyrosine kinase, released following AXL activation, demonstrated predictive value for Alzheimer’s disease development [33]. 

#### 4.2.2. Apolipoprotein E2 (ApoE2) 

ApoE, which exists in three allelic variants—*APOE*ε2, *APOE*ε3, and *APOE*ε4—encoding distinct isoforms, does not typically cross the blood–brain barrier. In the periphery, it is mainly synthesised by hepatocytes, which play a role in the elimination of triglyceride-rich lipoproteins, while in the central nervous system (CNS), its sources include astrocytes, microglia, vascular wall cells, and the choroid plexus, with stressed neurons contributing to a lesser extent. In the CNS, ApoE interacts with receptors that participate in processes such as lipid transport and Amyloid-β clearance, signal transduction, and intracellular trafficking of synaptic receptors. In addition, ApoE interacts with TREM2, influencing microglial phagocytosis of Amyloid-β and damaged neurons, while contributing to the maintenance of the neurodegenerative phenotype of disease-associated microglia. Regarding Alzheimer’s disease, ApoE2 has be shown to decrease its risk, even though the precise mechanisms are not yet clear [34]. While we were unable to instrument ApoE2 abundance in the plasma or gene expression in the brain and did not find any evidence for its causal role in any neuropsychiatric disorders, we were able to identify evidence of a shared causal variant between ApoE2 abundance in the CSF and brain and lower screen time duration (PPH4 = 0.99 and 0.94, respectively). This is a novel and interesting finding, noting the accumulating evidence for the role of screen time in the risk of psychiatric disorders [35].

#### 4.2.3. Low-Density Lipoprotein Receptor-Related Protein 8 (LRP8) 

LRP8, through multiple pathways and mechanisms, appears to play a role in both modulating neuronal activity and regulating cell proliferation. To summarise, it has been shown that higher levels of LRP8 in the periphery correlate with proliferation and metastasis in cancer cells and tissues, while low levels in the CNS appear to decrease functions related to neuronal migration, amyloidosis, and neurodegeneration, although the complexity of this remains to be fully explored [36]. We found that higher relative LRP8 abundance in the CSF and plasma, but lower cortical expression of *LRP8* was related to lower intelligence scores. These results are consistent with experimental studies showing impaired development of the neocortex [37] and cognitive decline [38] in mice with lower LRP8 expression in the brain.

In addition, higher cortical expression of *LRP8*, but lower relative abundance in the CSF and plasma was related to higher odds of engaging in physical activity and a higher liability to schizophrenia (LRP8_CSF_ and LRP8_cortex_ only) and BPD. Evidence for this protein’s involvement in neurological disorders is largely related to its role in Alzheimer’s disease [36,39,40], including supportive evidence of higher levels of LRP8 in Alzheimer’s disease cases vs. controls [41]. Regarding BPD, associations with gene expression in the brain tissue have differed across regions and specific LRP8 fractions [41]. Notably, available research relies almost exclusive on animal models or case-control epidemiological studies, therefore limiting the ability to infer potential causal effects in humans, specifically. Our novel findings, therefore highlight the importance of hypothesis-free studies for the identification of risk factors, and also the utility of MR in the investigation of complex diseases. 

#### 4.2.4. Hepatocyte Growth Factor-like Protein (MSP)

Higher genetically predicted levels of MSP in the CSF and brain were associated with lower fluid intelligence test scores, longer screen time, an increase in anorexia nervosa liability, and a lower likelihood of being physically active. Colocalization analysis identified the latter risk factor to likely share a causal variant with genetically predicted MSP in both tissues. Given its far-reaching biological impact as an inflammatory regulator [42], it was unsurprising that MSP, also called macrophage-stimulating protein, was one of the central nervous system proteins with the broadest effect on our outcomes of interest. 

### 4.3. Strengths and Limitations of This Study

The MR framework presents an efficient and economical approach to acquiring clinically significant insights into the impacts of proteomic exposures on diverse neuropsychiatric and behavioural traits. Using robust genetic markers as instruments for protein levels and expression, we circumvent some of the bias and limitations of traditional epidemiological studies, such as reverse causation. Indeed, the presence of neuropsychiatric disease or behavioural traits themselves might influence behaviours and biological pathways that subsequently alter the protein expression and abundance [2]. By including proteins across different tissue types, we were able to decipher underlying divergent or complementary biological processes and increase the robustness of our results [9]. 

Noteworthy limitations of our study are discussed below. When utilizing an aptamer-binding platform for proteomic profiling, recognition disparities, rather than actual protein abundance changes, could introduce bias [11]. In addition, genetic variants linked to proteins might not accurately proxy their functional consequences, potentially affecting specific isoforms while sparing others. Our study examines only a small fraction of the CSF/brain proteome, highlighting the extensive unexplored proteomic landscape. Another limitation of MR interpretation is that our observation on associations between proteins and neurological disorders might not be disease specific, as evidenced by multiple associations for a single protein depending also on how diseases are diagnosed and coded in the source data. Our analyses are only able to examine the potential effect of the investigated proteins on disease risk, but not progression. Therefore, our results are able to highlight the putative therapeutic targets that affect the risk (so that the targets could potentially be used for prevention), but not necessarily detect those that affect the progression [43]. In addition, further exploration is needed to understand and assess the biological mechanisms involved. Finally, our findings might not be generalizable to populations of non-European ancestries because of our reliance on mostly European ancestry-derived GWAS summary statistics.

## 5. Conclusions

In conclusion, this comprehensive MR study provides evidence for a causal role of tissue-specific protein abundance and expression in the risk of a range of neuropsychiatric disorders and risk factors. Although further validation and mechanistic exploration is required, our study provides valuable insights into the complex molecular underpinnings of neuropsychiatric disorders and holds promise for future advances in therapeutic opportunities.

## Figures and Tables

**Figure 1 biomedicines-12-00327-f001:**
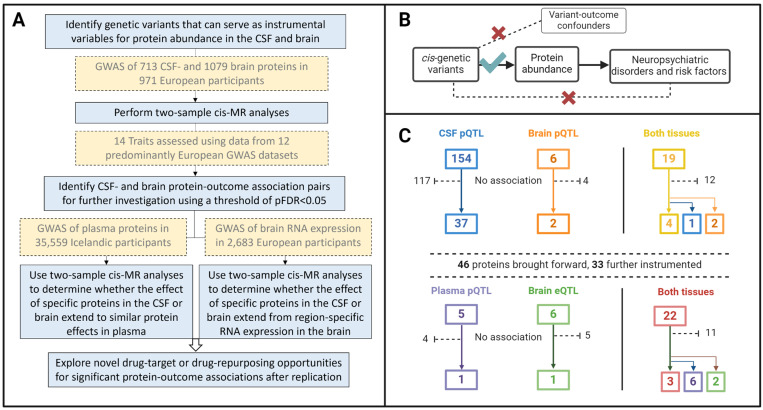
(**A**) Flow diagram of the study design and data sources. (**B**) The three instrumental variable assumptions of the Mendelian randomization framework made in this work: (1) The cis genetic variants variables are strongly associated to the relative abundance of the proteins they are instrumenting; (2) there are no instrument–outcome confounders, and (3) the genetic variants only affect the neuropsychiatric disorders and risk factors through the proteins they instrument, and not through any other independent causal pathway. (**C**) Overview of the number of proteins that were instrumented and observed to affect one or more outcomes in each tissue. CSF, cerebrospinal fluid; GWAS, genome-wide association study; FDR, false discovery rate; MR, Mendelian randomization. Created with BioRender.com.

**Figure 2 biomedicines-12-00327-f002:**
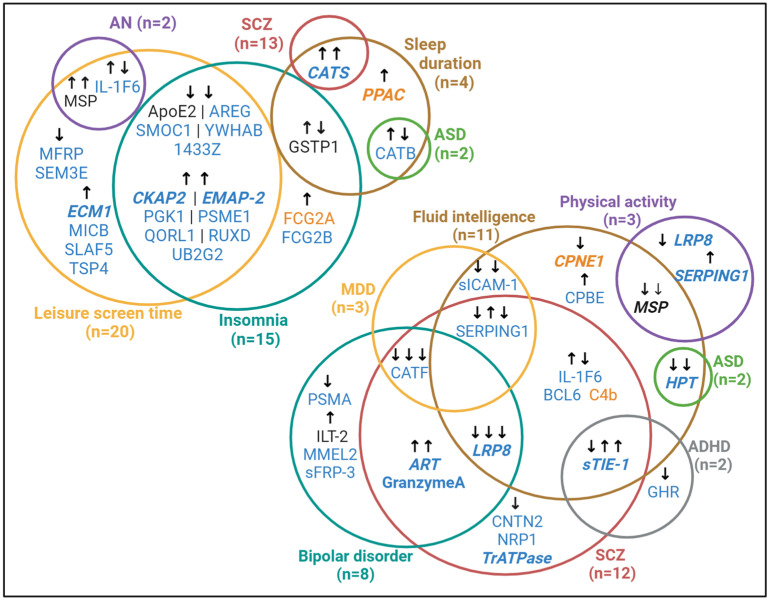
Overlap of effects of genetically predicted CSF and brain proteins on neuropsychiatric disorders and their risk factors. Protein name text colour indicates whether a statistically significant association was observed in genetically predicted CSF-derived (blue) proteins, brain-derived (orange) proteins, or both (black). Protein names written in bold, italicized text are those that were replicated in analysis of instrumented plasma abundance and/or brain expression. Arrows indicate the direction of association: arrows pointing up indicate a positive association (e.g, higher protein levels are associated with a higher disease liability or longer screen time) and those pointing down indicate a negative association (e.g, higher protein levels are associated with a lower disease liability or shorter screen time. Related association statistics are reported in Appendix A. ADHD, attention deficit hyperactivity disorder; AN, anorexia nervosa; ASD, autism spectrum disorder; MDD, major depressive disorder; SCZ, schizophrenia. Created with BioRender.com.

**Table 1 biomedicines-12-00327-t001:** Mendelian randomization estimates for the instrumented proteins that affect neuropsychiatric disorders or their risk factors in more than one of the investigated tissues.

Protein Full Name (Abbreviation)	Outcome (Unit of Estimate Measure)	CSF (cis-pQTL)		Brain (cis-pQTL)		Plasma (cis-pQTL)		Brain (cis-eQTL)	
		Estimate (95% CI)	pFDR	Estimate (95% CI)	pFDR	Estimate (95% CI)	pFDR	Estimate (95% CI)	pFDR
Apolipoprotein E2 (Apo E2)	Screen time (h)	−1.26 (−1.73; −0.79)	2 × 10^−5^	−0.31 (−0.42; −0.20)	4 × 10^−6^				
	Insomnia (OR)	0.41 (0.22; 0.77)	0.09	0.79 (0.68; 0.91)	0.03				
Agouti-related protein (ART)	Schizophrenia (OR)	2.14 (1.4; 3.26)	0.02			1.09 (1.03; 1.16)	0.02		
Cathepsin S (CATS)	Schizophrenia (OR)	1.64 (1.27; 2.13)	0.008			1.14 (1.06; 1.22)	0.002		
	Sleep duration (h)	0.11 (0.05; 0.17)	0.03			0.02 (0.01;0.03)	0.02		
Cytoskeleton-associated protein 2 (CKAP2)	Screen time (h)	0.71 (0.46; 0.96)	5 × 10^−6^					SC: 0.01 (0.01; 0.02)	0.001
								HC: 0.01 (0.00; 0.02)	0.01
Copine 1 (CPNE1)	Intelligence (points)			−0.08 (−0.12; 0.04)	0.002			SC: −0.01 (−0.02; −0.01)	9 × 10^−4^
								HC: −0.02 (−0.03; −0.01)	0.002
Extracellular matrix protein 1 (ECM1)	Screen time (h)	0.67 (0.37; 0.97)	7 × 10^−4^					0.05 (0.03; 0.07)	4 × 10^−4^
Glutathione S-transferase P (GSTP1)	Insomnia (OR)	1.18 (1.06; 1.31)	0.05	1.44 (1.14; 1.82)	0.04				
	Sleep duration (h)	−0.07 (−0.12; −0.03)	0.04	−0.16 (−0.27; −0.06)	0.04				
Haptoglobin (HPT)	ASD (OR)	0.92 (0.88; 0.97)	0.03					1.05 (1.01; 1.09)	0.04
Leukocyte immunoglobulin-like receptor subfamily B member 1 (ILT-2)	Bipolar disorder (OR)	1.33 (1.11; 1.59)	0.04	1.63 (1.19; 2.23)	0.04				
Low-density lipoprotein receptor-related protein 8 (LRP8)	Bipolar disorder (OR)	0.45 (0.31; 0.65)	0.001			0.84 (0.77; 0.93)	0.004	1.19 (1.1; 1.3)	4 × 10^−4^
	Schizophrenia (OR)	0.57 (0.42; 0.76)	0.008					1.11 (1.03; 1.19)	0.02
	Physical activity (OR)	0.81 (0.71; 0.91)	0.02			0.95 (0.91; 0.98)	0.01	1.06 (1.02; 1.1)	0.01
	Intelligence (points)	−0.21 (−0.32; −0.1)	0.008			−0.05 (−0.08; −0.02)	0.01	0.05 (0.02; 0.07)	2 × 10^−4^
Hepatocytes growth factor-like protein (MSP)	Intelligence (points)	−0.11 (−0.13; −0.09)	9 × 10^−18^	−0.21 (−0.25; −0.16)	4 × 10^−19^	−0.02 (−0.03; −0.01)	6 × 10^−5^	0.04 (0.02; 0.05)	1 × 10^−5^
	Anorexia nervosa (OR)	1.30 (1.17; 1.45)	1 × 10^−4^	1.61 (1.32; 1.97)	0.04				
	Physical activity (OR)	0.95 (0.92; 0.97)	8 × 10^−4^	0.90 (0.86; 0.95)	8 × 10^−4^				
	Sleep duration (hs)	0.08 (0.06; 0.11)	2 × 10^−6^	0.16 (0.11; 0.21)	1 × 10^−7^				
Plasma protease C1 inhibitor (SERPING1)	MDD (OR)	0.72 (0.6; 0.85)	0.006			0.97 (0.94; 0.99)	0.02		
	Schizophrenia (OR)	0.46 (0.34; 0.63)	1 × 10^−4^			0.93 (0.9; 0.97)	8 × 10^−3^		
	Physical activity (OR)	1.36 (1.2; 1.54)	1 × 10^−4^			1.03 (1.02; 1.06)	6 × 10^−5^		
Tyrosine-protein kinase receptor Tie-1 (s-Tie1)	ADHD (OR)	12.1 (4.73; 30.8)	3 × 10^−5^			1.24 (1.05; 1.45)	0.04	1.15 (1.1; 1.22)	5 × 10^−6^
	Schizophrenia (OR)	2.84 (1.64; 4.93)	0.009			1.11 (1.05; 1.16)	0.002	1.06 (1.02; 1.09)	0.004
	Intelligence (points)	−0.43 (−0.62; −0.23)	0.001					−0.21 (−0.32; −0.1)	3 × 10^−4^
LMW phosphotyrosine protein phosphatase (PPAC)	Sleep duration (h)			0.04 (0.01; 0.06)	0.04			−0.03 (−0.05; −0.01)	0.01
Tartrate-resistant acid phosphatase type 5 (TrATPase)	Schizophrenia (OR)	0.55 (0.38; 0.79)	0.03					0.92 (0.88; 0.97)	0.01
Thrombospondin-4 (TSP4)	Screen time (h)	0.34 (0.17; 0.52)	0.008					0.04 (0.02; 0.07)	0.007

Genetically predicted protein expression in the brain (Brain eQTLs) refer to expression in the cortex unless specified as hippocampus (HC) or spinal cord (SC). ADHD, attention deficit hyperactivity disorder; ASD, autism spectrum disorder; h, hours; HC, hippocampus; MDD, major depressive disorder; OR, odds ratio; SC, spinal cord.

**Table 2 biomedicines-12-00327-t002:** Colocalization results for the significant genetic associations between instrumented proteins on neuropsychiatric disorders or their risk factors.

					Posterior Probability of Causal Variant Hypotheses (PPH)				
Uniprot	Tissue	Exposure	Outcome	SNPs	0: None	1: Exposure	2: Outcome	3: Distinct	4: Both
P02649	CSF	ApoE2	Insomnia	247	<0.01	0.70	<0.01	<0.01	0.30
	**CSF**	**ApoE2**	**Leisure screen time**	253	<0.01	<0.01	<0.01	0.01	**0.99**
	**Brain**	**ApoE2**	**Leisure screen time**	232	<0.01	<0.01	<0.01	0.06	**0.94**
Q14114	**CSF**	**LRP8**	**Bipolar disorder**	356	<0.01	0.05	<0.01	0.04	**0.91**
	**Plasma**	**LRP8**	**Bipolar disorder**	5038	<0.01	0.04	<0.01	0.05	**0.91**
	**Cortex**	**LRP8**	**Bipolar disorder**	4975	<0.01	0.04	<0.01	0.05	**0.91**
	**CSF**	**LRP8**	**Fluid intelligence**	376	<0.01	0.27	<0.01	0.04	**0.70**
	**Plasma**	**LRP8**	**Fluid intelligence**	6868	<0.01	0.36	<0.01	0.10	**0.54**
	**Cortex**	**LRP8**	**Fluid intelligence**	5656	<0.01	0.33	<0.01	0.07	**0.59**
	CSF	LRP8	Schizophrenia	356	<0.01	0.50	<0.01	0.02	0.48
	Cortex	LRP8	Schizophrenia	4999	<0.01	0.47	<0.01	0.10	0.43
	CSF	LRP8	Physical activity	393	<0.01	0.47	<0.01	0.03	**0.51**
	Plasma	LRP8	Physical Activity	8558	<0.01	0.60	<0.01	0.31	0.09
	Cortex	LRP8	Physical Activity	5969	<0.01	0.44	<0.01	0.14	0.42
P26927	CSF	MSP	Anorexia Nervosa	219	<0.01	<0.01	<0.01	>0.99	<0.01
	Brain	MSP	Anorexia Nervosa	222	<0.01	<0.01	<0.01	>0.99	<0.01
	CSF	MSP	Fluid intelligence	219	<0.01	<0.01	<0.01	>0.99	<0.01
	Brain	MSP	Fluid intelligence	222	<0.01	<0.01	<0.01	>0.99	<0.01
	Plasma	MSP	Fluid intelligence	3558	<0.01	<0.01	<0.01	>0.99	<0.01
	Cortex	MSP	Fluid intelligence	2792	<0.01	<0.01	<0.01	>0.99	<0.01
	CSF	MSP	Sleep duration	219	<0.01	0.97	<0.01	0.02	<0.01
	Brain	MSP	Sleep duration	222	<0.01	0.97	<0.01	0.02	0.01
	**CSF**	**MSP**	**Physical activity**	236	<0.01	<0.01	<0.01	0.05	**0.94**
	**Brain**	**MSP**	**Physical activity**	229	<0.01	<0.01	<0.01	0.03	**0.97**
P35590	**CSF**	**sTie-1**	**ADHD**	241	<0.01	<0.01	<0.01	0.05	**0.95**
	Plasma	sTie-1	ADHD	4474	<0.01	<0.01	<0.01	1.00	<0.01
	Cortex	sTie-1	ADHD	4219	<0.01	<0.01	<0.01	1.00	<0.01
	**CSF**	**sTie-1**	**Schizophrenia**	234	<0.01	0.05	<0.01	0.05	**0.90**
	Plasma	sTie-1	Schizophrenia	4617	<0.01	<0.01	<0.01	1.00	<0.01
	Cortex	sTie-1	Schizophrenia	4581	<0.01	<0.01	<0.01	1.00	<0.01
	**CSF**	**sTie-1**	**Fluid intelligence**	234	<0.01	<0.01	<0.01	0.05	**0.95**
	Cortex	sTie-1	Fluid intelligence	4888	<0.01	<0.01	<0.01	1.00	<0.01

Posterior probabilities of the following hypotheses are tested: 0: no variants are causal; 1: causal variant for exposure only; 2: causal variant for outcome only; 3: distinct causal variants for exposure and outcome; 4: shared causal variant for exposure and outcome. **Bold text** indicates exposure–outcome pairs that likely share a causal variant. CSF, Cerebrospinal fluid; ADHD, attention deficit hyperactivity disorder.

## Data Availability

All data used in this study are publicly available from the cited sources.

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
