# Peer review of "Cerebrospinal and Brain Proteins Implicated in Neuropsychiatric and Risk Factor Traits: Evidence from Mendelian Randomization"

_biomedicines, 2024, doi:10.3390/biomedicines12020327_

Round 1
Reviewer 1 Report
Comments and Suggestions for Authors
This analyses (MR) are only able to examine the effect on the disease risk, but not progression.
1 - This research provides genetics evidence supporting novel tissue-specific proteomic targets for neuropsychiatric disorders and their risk factors.
2 - Further exploration is necessary to understand biological mechanisms and assess.
3. The comprehensive Mendelian randomization (MR) revealed evidence for a casual role of tissue-specific protein abundance and expression in the risk of neuropsychiatric disorders.
4. The conclusions are consistent with the results.
5. This study provides valuable information about the relationship between proteins and neurological disorders. These analyzes examine the effect on the risk of developing the disease but not on its progression.
6. The References selected appropriately.
7. No comments for tables and figures.
Author Response
Reviewer 1
This analyses (MR) are only able to examine the effect on the disease risk, but not progression.
We thank the Reviewer for their expert feedback. This statement was indeed highlighted in our limitation section in the Discussion section lines 423-427: “Our analyses are only able to examine the effect on the disease risk, but not progression. Therefore, our results are able to highlight the putative therapeutic targets that affect the risk (so that the targets could potentially be used for prevention), but not necessarily detect those that affect the progression”.
1 - This research provides genetics evidence supporting novel tissue-specific proteomic targets for neuropsychiatric disorders and their risk factors.
We thank the Reviewer for this concise summary of our work. We now detail this in lines 299-300 in the Discussion section:
“This research provides genetic evidence in support of novel tissue-specific protein and gene targets for neuropsychiatric disorders and their risk factors.”
2 - Further exploration is necessary to understand biological mechanisms and assess.
We acknowledge this limitation, and now detail it in lines 437-428 in the Discussion section:
“In addition, further exploration is needed to understand and assess the biological mechanisms involved.”
- The comprehensive Mendelian randomization (MR) revealed evidence for a casual role of tissue-specific protein abundance and expression in the risk of neuropsychiatric disorders.
We thank the Reviewer again for this accurate summary, which is now incorporated into our Discussion section in lines 299-300:
“This research provides genetic evidence in support of novel tissue-specific protein and gene targets for neuropsychiatric disorders and their risk factors.”
- The conclusions are consistent with the results.
We thank the Reviewer for their positive feedback.
- This study provides valuable information about the relationship between proteins and neurological disorders. These analyzes examine the effect on the risk of developing the disease but not on its progression.
We thank the Reviewer for their expert feedback. This was effectively acknowledged in the limitations section lines 423-427.
- The References selected appropriately.
We thank the Reviewer for their positive feedback.
- No comments for tables and figures.
We thank the Reviewer for their positive feedback.
Reviewer 2 Report
Comments and Suggestions for Authors
Both the Results and Discussion sections of the paper could be improved, the text is a bit dense. In the Discussion more information about other functions of the described genes could be discussed, i.e., other than effects on brain functioning (e.g., ApoE plays many roles in the body). In the text (Results) other brain areas (e.g., hippocampus) are mentioned, but it is unclear inn how many patients this has been tested, these numbers have to be much lower than brain level testing.
Comments on the Quality of English LanguageText is a bit dense in the results section.
Author Response
- Both the Results and Discussion sections of the paper could be improved, the text is a bit dense.
We thank the Reviewer for their comment, which helped to make our manuscript more readable. To make the results and discussion section clearer and easier, headings and sub-headings have been added as follow:
- Results
3.1 CSF and brain genetic instruments and Two-Sample MR associations
3.2 Protein-specific and outcome-specific associations in CSF/brain sample
3.3. Plasma and brain gene expression genetic instruments and Two-Sample MR associations
3.4 Evidence of shared causal variants in concordant specific-tissue protein-outcome associations
- Discussion
4.1. Genetic exploration of neuropsychiatric disorders: unraveling causal tissue-specific protein associations
4.2 Focus on specific proteins: insights into sTie-1, LRP8, ApoE2, and MSP
4.3. Strenghts and limitations of this study
Some long sentences have been limited as much as possible to preserve the meaning of our results and discussion. These modifications have been highlighted in the main text.
- In the Discussion more information about other functions of the described genes could be discussed, i.e., other than effects on brain functioning (e.g., ApoE plays many roles in the body).
We thank the Reviewer for this precious comment. Regarding ApoE2, a paragraph explaining more in deep its function has been provided in lines 353-364:
“ApoE, which exists in three allelic variants - APOEε2, APOEε3 and APOEε4 - en-coding distinct isoforms, does not cross the blood-brain barrier. In the periphery, it is mainly synthesised by hepatocytes, which play a role in the elimination of triglyceride-rich lipoproteins, while in the central nervous system (CNS), its sources include astrocytes, microglia, vascular wall cells and the choroid plexus, with stressed neurons con-tributing to a lesser extent. In the CNS, APOE interacts with receptors such as LDLR, VLDLR, HSPG, LRP1 and LRP8, participating in processes such as lipid transport and Aβ clearance, signal transduction and intracellular trafficking of synaptic receptors. In addition, ApoE interacts with TREM2, influencing microglial phagocytosis of Aβ and damaged neurons, while contributing to the maintenance of the neurodegenerative phenotype of disease-associated microglia. Regarding Alzheimer's disease, ApoE2 has be shown to decrease its risk, even though the precise mechanisms are not yet clear.”
Similarly, we improved the explanation of the LRP8’s functions in a paragraph in lines 374-386:
“LRP8, through multiple pathways and mechanisms, appears to play a role in both modulating neuronal activity and regulating cell proliferation. To oversimplify, it has been shown that high levels of LRP8 in the periphery correlate with proliferation and metastasis in cancer cells and tissues, while low levels in the CNS appear to decrease functions related to neuronal migration, amyloidosis and neurodegeneration, although the complexity of this is not fully defined.LRP8, also known as apolipoprotein E recep-tor 2, acts as a receptor for Reelin–a regulator of neuronal migration–and apolipopro-tein E, a primary risk factor in the development of Alzheimer’s disease [36]. We found that higher cortical expression of LRP8, but lower relative abundance in the CSF and plasma, but higher cortical expression of LRP8, which could be the result of negative feedback loop, was related to lower intelligence scores. These results are consistent with experimental studies showing impaired development of the neocortex and cognitive decline in LRPS deficiency in mice.”
- In the text (Results) other brain areas (e.g., hippocampus) are mentioned, but it is unclear in how many patients this has been tested, these numbers have to be much lower than brain level testing.
We thank the Reviewer for their expert feedback. Indeed, the sample size to derived SNPs for specific-protein gene expression level in hippocampus region was lower than in the cortex region. We now detail it in the Methodology section 2.2.4 lines 169-170 :
“Genetic associations with gene expression levels in cortex, hippocampus, and spinal cord were obtained from a meta-analysis of 14 cohorts, consisting of up to 2,683, 168 and, 108 European ancestry individuals, respectively.”
- Text is a bit dense in the results section.
We thank the Reviewer for their comment, which helped to make our manuscript more readable. We provide text modifications as shown in comment 1.